# Reviewing the Effects of Skin Manipulations on Adult Newt Limb Regeneration: Implications for the Subcutaneous Origin of Axial Pattern Formation

**DOI:** 10.3390/biomedicines9101426

**Published:** 2021-10-09

**Authors:** Martin Miguel Casco-Robles, Kayo Yasuda, Kensuke Yahata, Fumiaki Maruo, Chikafumi Chiba

**Affiliations:** 1Faculty of Life and Environmental Sciences, University of Tsukuba, Tennodai 1-1-1, Tsukuba 305-8572, Ibaraki, Japan; casco.miguel.gm@u.tsukuba.ac.jp (M.M.C.-R.); yahata@biol.tsukuba.ac.jp (K.Y.); maru@biol.tsukuba.ac.jp (F.M.); 2Graduate School of Life and Environmental Sciences, University of Tsukuba, Tennodai 1-1-1, Tsukuba 305-8572, Ibaraki, Japan; s2020952@u.tsukuba.ac.jp

**Keywords:** newt, limb regeneration, skin, pattern formation

## Abstract

Newts are unique salamanders that can regenerate their limbs as postmetamorphic adults. In order to regenerate human limbs as newts do, it is necessary to determine whether the cells homologous to those contributing to the limb regeneration of adult newts also exist in humans. Previous skin manipulation studies in larval amphibians have suggested that stump skin plays a pivotal role in the axial patterning of regenerating limbs. However, in adult newts such studies are limited, though they are informative. Therefore, in this article we have conducted skin manipulation experiments such as rotating the skin 180° around the proximodistal axis of the limb and replacing half of the skin with that of another location on the limb or body. We found that, contrary to our expectations, adult newts robustly regenerated limbs with a normal axial pattern regardless of skin manipulation, and that the appearance of abnormalities was stochastic. Our results suggest that the tissue under the skin, rather than the skin itself, in the intact limb is of primary importance in ensuring the normal axial pattern formation in adult newt limb regeneration. We propose that the important tissues are located in small areas underlying the ventral anterior and ventral posterior skin.

## 1. Introduction

The newt, a species of salamander, serves as a model for limb regeneration in adults [1,2,3]. Salamanders are in general able to regenerate limbs in the larval stage. Larval limb regeneration is based on stem/progenitor cells except for cartilage/bone, which is partially regenerated from fibroblasts/interstitial cells [4]. However, in most salamanders such regenerative capacity declines following metamorphosis, during which the body’s system is drastically remodeled to adapt to a terrestrial environment. On the other hand, newts preserve their high regenerative capacity even as they grow beyond metamorphosis by switching their cellular mechanism of regeneration from a stem/progenitor-based mode to a dedifferentiation-based mode wherein fully differentiated cells, such as contractile muscle fibers, can be recruited as an alternative to declined stem/progenitor cells [1]. Furthermore, the mechanism of digit pattern formation in forelimb regeneration is switched from an aquatic mode (preaxially dominant digit formation) to a terrestrial mode (synchronized digit formation) during metamorphosis [5]. Thus, although the process of limb regeneration in adult newts resembles that in larvae, the cellular participants and the mechanisms of pattern formation cannot all be the same.

When the limb of an adult newt is amputated, the wound is covered with an epidermis, referred to as the ‘wound epidermis’, which extends from the wound edge of the skin around the stump, and then a mass of mesenchymal cells, referred to as the ‘blastema’, forms under the epidermis [1,2,3]. The mesenchymal cells of the blastema originate from various tissues in the stump. During the formation of the blastema, severed nerves and blood capillaries at the stump also grow into the blastema [6,7]. The blastema establishes a three-dimensional axial pattern from which a patterned limb is eventually regenerated [2]. As an analogy of limb bud development, the axial patterning of the blastema is thought to be achieved by the interaction between cells in the blastema and the epidermis surrounding the blastema [2,3]. In order to regenerate human limbs as newts do, it is therefore necessary to determine whether the cells homologous to those contributing to the axial patterning of the blastema in newts also exist in humans. However, it is not yet clear which cell types play this role in newts.

In studies of amphibians, the accumulated evidence indicates that the capacity of the blastema to regenerate the limb depends on their level along the proximodistal axis of the limb, thus allowing the blastema to accurately regenerate a missing distal part of the limb from the stump at any level [2,3]. Therefore, the blastema is believed to have a positional identity/memory. The evidence further suggests that at any level along the proximodistal axis of the limb, the skin surrounding the stump plays, in combination with nerves, a pivotal role in growth and axial patterning of the blastema [2,3,6,8]. Amphibian skin is essentially composed of the epidermis (epithelial layer) and the dermis (mesenchyme) which are separated by a pigment cell layer [1]. The epidermis is, as mentioned above, the origin of the wound epidermis which eventually forms the epidermis of the skin of the regenerated limb [1,4]. Mesenchymal cells arising from the dermis also contribute to the blastema, which eventually forms the dermis itself of the skin of the regenerated limb and becomes a part of the cartilage/bone of the regenerated limb [1,4].

In adult newts, with respect to the proximodistal patterning of regenerating limbs, a Prod 1–nAG signaling system is known to be involved in establishing the positional identity of the blastema [6,9,10]. Prod 1 is a three-finger protein that is attached to the cell surface with a glycosylphosphatidylinositol (GPI) anchor, and in the intact limb is expressed with a proximodistal gradient (proximal > distal). During limb regeneration, Prod 1 is uniformly expressed in the mesenchymal cells in the early blastema and not in the wound epidermis, although the expression intensity in the blastema is different between the levels at which the blastema is formed along the proximodistal axis (proximal > distal) [9,10]. nAG, a newt anterior gradient protein, is a secreted ligand for Prod 1 and a growth factor for blastemal cells. During limb regeneration, nAG is expressed in the regenerating nerve in the blastema and the wound epidermis surrounding the top of the blastema. nAG expression in the wound epidermis strongly depends on the presence of the regenerating nerve and is required for the blastema to develop into a patterned limb [6,10]. Thus, for the proximodistal patterning and growth of the blastema, the interaction between blastemal cells with positional identity and the regenerating nerve/wound epidermis is necessary.

On the other hand, the mechanism of the anteroposterior and dorsoventral patterning of the regenerating limbs is still unclear. Moreover, the evidence accumulated in the axolotl, a paedomorphic salamander, suggests that the way the skin acts on the patterning of the blastema may be more complex. A recent study has demonstrated that the interactions between mesenchymal cells originating from anterior and posterior skin of the limb in the presence of nerves on the wound epidermis during blastema growth are necessary and sufficient for complete regeneration of a patterned limb [2,3,8,11]. This finding suggests that if the skin on the anterior half of the limb is replaced with skin obtained from the posterior half of the contralateral limb, or vice versa, while keeping the other two axes (i.e., proximodistal and dorsoventral axes) of the skin unchanged, the limb should lose its regenerative ability. However, there are reports indicating that such skin manipulation does not suppress limb regeneration itself, but rather gives rise to the supernumerary digit (excess-fingered) limb [2,12]. Based on the results in a series of skin manipulation studies, it is suggested that dermal fibroblasts in posterior and dorsal skin are responsible for most of the regeneration potential of the limb [2,12].

In adult newts, there are few studies that performed skin manipulation to address how the skin around the stump is involved in the axial patterning of the blastema and the regenerating limbs [13,14,15]. Therefore, in this study, using the forelimbs of the adult newt *Cynops pyrrhogaster*, we conducted skin manipulation experiments such as rotating the skin 180° around the proximodistal axis of the limb (hereafter termed ‘180° skin rotation’) and replaced half of the skin with that of another location on the limb or body (hereafter termed ‘half skin graft operation’). We report that adult newts robustly regenerated limbs with a normal axial pattern regardless of the skin manipulation, and that the appearance of abnormalities was stochastic. Our results lead to a hypothesis that in the adult newt, those cells that primarily contribute to the axial patterning of the blastema may reside subcutaneously rather than in the skin.

## 2. Materials and Methods

All methods were carried out in accordance with the Regulations on the Handling of Animal Experiments in the University of Tsukuba. All experimental protocols were approved by the University of Tsukuba Safety Committee for Recombinant DNA Experiments (Code: 170110) in which the policy of the Animal Care and Use Committee in the University of Tsukuba was included. Moreover, all methods were performed in accordance with the ARRIVE guidelines.

### 2.1. Animals

The Japanese fire-bellied newt *C. pyrrhogaster* at the adult stage (total body length: males, ~9 cm; females, 11–12 cm) was used in this study. The animals were captured from Niigata, Ishikawa, Aichi, and Chiba Prefectures by a supplier (Aqua Grace, Yokohama, Japan) and reared in plastic containers at 18 °C under natural light conditions. Animals were fed daily with frozen mosquito larvae (Akamushi; Kyorin Co., Ltd., Hyogo, Japan) and the containers were kept clean [16,17].

### 2.2. Anesthesia

An anesthetic, FA100 (4-allyl-2-methoxyphenol; DS Pharma Animal Health, Osaka, Japan) dissolved in tap water (*v*/*v*) was used at 18 °C [16,17]. The animals were placed in 0.1% FA100 solution (5 or less individuals/300 mL) and anesthetized for the following periods of time: for skin manipulation, 2.5 h; for limb amputation, 30 min; for taking pictures of the animals, 1 h.

### 2.3. Skin Manipulation

Following anesthesia, the animals were rinsed with filtered tap water and dried on a paper towel (Elleair Prowipe, Soft High Towel, Unbleached, 4P; Dio Paper Corporation, Tokyo, Japan), and then transferred to a silicon bottom chamber (L × W × H (cm): 15.5 × 8 × 2.8; silicon, 1 cm in height) placed on a dissecting microscope (SZX7 and SZ61; Olympus, Tokyo, Japan; M165 FC; Leica Microsystems, Wetzlar, Germany).

For the 180° skin rotation, the skin (3–4 mm long along the proximodistal axis) surrounding the upper arm of the right forelimb was carefully peeled off by manipulating fine surgical forceps and blade from a slit which was first made on the dorsal surface of the skin. The excised skin was spread on a clean sheet of paper (Elleair Prowipe, Soft Wiper S200; Dio Paper Corporation, Tokyo, Japan) humidified with phosphate-buffered saline solution (PBS, pH 7.5) and immediately placed back to the original site so that the skin was rotated 180° around the proximodistal axis of the limb (for details, see Figure 1). The operated animals were placed in a Tupperware box (W: 14.1 cm, D: 21.4 cm, H: 4 cm; one animal per box) containing crumpled pieces of half-dried paper towel (Elleair Prowipe, Soft High Towel, Unbleached, 4P; Dio Paper Corporation, Tokyo, Japan) and allowed to recover at 4 °C for 24 h. During the incubation at 4 °C, the animals were under anesthesia and did not move, so the grafted skin adhered to the limb without shedding. Then, the animals were reared in the same box at 18 °C for one month during which the gap of skin closed and the grafted skin became firmly attached to the limb. The moist boxes were cleaned every day. Feeding was restarted from 2 weeks later. At one month after the operation, the limb was amputated across the grafted skin (see below). As controls, the excised skin was placed back to the original site without rotation (sham surgery), or not so that the subcutaneous tissue was exposed (skin removal).

For the half skin graft operation, half skin (3–4 mm long along the proximodistal axis) on either the anterior, posterior, dorsal, or ventral side of the upper arm of the right forelimb was replaced with skin obtained from the contralateral upper arm or the flank/tail of the same individual by using basically the same procedures as those used for 180° skin rotation. Similarly, the operated animals were reared for one month and then the limb was amputated across the grafted skin (see below). As controls, the excised skin was placed back to the original site immediately after having been peeled off (sham surgery) or the limb was amputated immediately after the half skin was replaced (immediate amputation). Note that a skin allograft between adult individuals was impossible because the grafted skin dissolved and disappeared, probably due to immunological rejection by the host [18].

The grafted skin was easily distinguished from the host skin by its characteristic color pattern. We frequently observed the grafted skin under a dissecting microscope during the process of rearing animals after skin grafting or amputation. The grafted skin maintained its characteristics on the host limb for more than a year. However, if the grafted skin was detached or lost, the animal was excluded from the experiment (this case was less than 5% on average).

### 2.4. Limb Amputation

In the case of the skin removal option, the upper arm was amputated across the wound bed immediately after skin removal. In the case of the immediate amputation option, the upper arm was amputated along the gap between the host skin and the distal margin of the half skin immediately after the half skin was placed on the limb (the half skin was not cut again to prevent its detachment or loss). In both cases, animals were still under anesthesia. In other cases, the animals (one month after operation) were anesthetized in 0.1% FA100 solution for 30 min prior to amputation, rinsed with filtered tap water, and dried on a paper towel. In all cases, the upper arm was amputated under a dissecting microscope (M165 FC; Leica Microsystems) using a carbon steel blade (FA-10, Double Edge Blade, Feather Safety Razor, Osaka, Japan). Amputees were placed back in the Tupperware box containing a dry paper towel, allowed to recover at 4 °C for 24 h (for the skin removal option and the immediate amputation option) or 5 h (for the standard skin grafting options), and then, after blood was removed from the body surface, reared in the same box containing crumpled pieces of half-dried paper towels at 18 °C. Empirically, the crumpled paper towel was essential to ensure limb regeneration, as it allowed the animal to rest on the folds while lifting the injured forelimb. The moist boxes were cleaned every other day. Feeding was restarted by hand as soon as the wound was completely covered by the epidermis. The operated animals were never placed in water to prevent the regenerating tissue from damage/infection. Note that in this study we did not count the animals that obviously suffered damage on the grafted skin from amputation or on the blastema during rearing (this case was less than 5% on average).

### 2.5. Skeletal Staining of Limbs

Intact and regenerating limbs were collected from anesthetized animals by amputation at the midpoint of the upper arm. At 22 °C, the limb samples were fixed in 4% paraformaldehyde in PBS for 15 h, rinsed in PBS containing 0.1% Tween20 (PBST) three times for 1 h each, and then allowed to stand in fresh PBST for 15 h; transferred to 3% acetic acid aqueous solution, allowed to stand for 15 h, and then rinsed in PBST six times for 30 min each (3 h in total); transferred to an ethanol series (ethyl alcohol dissolved in MilliQ water; 25%, 50%, 75%, 90%, 100% (each for 1 h), and 100% (15 h)) to dehydrate them; stained in an Alcian blue solution (0.1% (*w*/*v*) Alcian Blue 8GX (Sigma-Aldrich Japan K.K., Tokyo, Japan) dissolved in a mixture of 70% ethyl alcohol and 30% acetic acid) for 2 days, and thoroughly rinsed in 100% ethyl alcohol until acetic acid was removed (5 h in total); transferred to another ethanol series (ethyl alcohol dissolved in MilliQ water; 100%, 90%, 75%, 50%, 25%, each for 1 h) to rehydrate, and rinsed in PBST three times for 1 h; bleached in 3% hydrogen peroxide solution containing 9% (*w*/*v*) KOH for 5 h, and thoroughly rinsed in PBST (3 h in total); incubated in PBS containing 1% (*w*/*v*) trypsin (TRYPSIN, 1-250; FUJIFILM Wako Pure Chemical Co., Ltd., Osaka, Japan) and 30% (*w*/*v*) sodium tetraborate (28-4440-5; Sigma-Aldrich Japan K.K.) at 37 °C until they became transparent (24–72 h); transferred to 0.5% (*w*/*v*) NaOH solution and incubated for 3 h; stained in 1% (*w*/*v*) Alizarin red (Alizarin Red S, A5533, Sigma-Aldrich Japan K.K.) dissolved in 0.5% (*w*/*v*) NaOH solution until bone became red (3–5 h), and thoroughly rinsed in 0.5% NaOH solution and PBST for 3 h each; transferred to glycerol series (glycerol dissolved in MilliQ water; 25%, 50%, 75%, 100% (each until the sample sank), and 100% ×2 (each for 24 h)), and stored at 4 °C until the skeletal patterns were examined.

The digits were identified by counting the number of cartilages/bones or joints (digit 1, 2; digit 2, 3; digit 3, 4; digit 4, 3). For the second and third digits, which have three cartilages/bones or joints, their relative length and position were also taken into account.

### 2.6. Image Acquisition and Data Analysis

A dissecting microscope (M165 FC; Leica Microsystems, Wetzlar, Germany) was used to monitor limb regeneration in living newts and to take images of those limbs whose skeletons were stained. Images or videos were taken while changing the focal plane with a digital camera (C-5060; Olympus, Tokyo, Japan) attached to the microscope and stored in a computer. Images were analyzed by Adobe Photoshop 2021 and with software for the image acquisition system. Figures were prepared using Adobe Photoshop 2021 (Adobe Inc., 345 Park Avenue, San Jose, CA, USA). Image brightness, contrast, and sharpness were adjusted according to the journal’s guidelines. Statistical analysis was performed using Ekuseru-Toukei software (v. 3.21, Social Survey Research Information, Tokyo, Japan).

## 3. Results

### 3.1. 180° Skin Rotation

If some of the mesenchymal cells of the blastema and the epidermis surrounding the blastema, derived from the skin at a particular location in the limb, provide the surrounding blastemal cells with a positional cue linked to their original location, alteration of the geometrical identity of the skin on the three-dimensional coordinates of the limb prior to amputation should have a profound effect on limb morphogenesis during regeneration. In this study, we first examined this hypothesis by rotating the skin of the upper arm (stylopod) 180° around the proximodistal axis (Figure 1). We amputated the upper arm across the rotated skin one month after rotation by which time the skin was fully engrafted, and then monitored the morphological changes of the regenerating part of the limb. As a result, contrary to the hypothesis, most of the operated limbs (~77%, 17/22) regenerated normally (Figure 1; Table 1).

In this surgical operation, we occasionally observed abnormalities in the regenerated limbs (~23%, 5/22; Figure 2; Table 1). In three cases (~14%), the regenerated limb was characteristically rotated 90° forward around the proximodistal axis, but the digit alignment of the hand (autopod) was reversed (i.e., with the first digit pointing up). In the other cases, small digits formed on the anterior side of the back of the hand (~5%, n = 1) or two digits formed on the top of the lower arm (~5%, n = 1). However, in either the sham surgery in which the excised skin was grafted back to its original position immediately after the excision, or the skin removal operation in which the limb was amputated immediately after the skin was peeled off (i.e., not grafted back), all limbs regenerated normally (n = 3 each; Table 1). Note that in the skin removal operation, the formation of the blastema did not start until the whole surface of the wound was completely covered with the epidermis. This indicates that covering the wound with epidermis (wound closure) is a necessary condition for the initiation of the regeneration. Moreover, this shows that the position of the skin from which the wound epidermis originates does not affect the positional identity or the axial patterning of the blastema because the wound edge of the skin is located more proximal to the tip of the stump.

Taken together, it appeared that, at least in the forelimbs of the adult newts, the geometrical identity of the skin was not essential for the patterning of the regenerated limb along the anteroposterior and dorsoventral axes. According to the results of the sham surgery and skin removal operations, the morphological abnormalities of the regenerated limb, which occurred at low probabilities, did not seem to be caused by the surgery itself but instead were related to the operation of rotating the skin.

### 3.2. Half Skin Graft Operation

Next, we investigated whether the interaction of the adjacent skin, such as the skin covering the anterior and posterior surfaces of the limb, was essential for the regeneration of the patterned limb. However, contrary to our expectations, most of the operated limbs regenerated normally (Table 2).

#### 3.2.1. Anterior-Anterior (An-An)

The skin on the posterior half of the upper arm was replaced with skin obtained from the anterior half of the contralateral upper arm while keeping the proximodistal and dorsoventral axes of the skin unchanged (Figure 3). The operated limbs regenerated normally (~81%, 13/16) except for the rare occasions in which a 5-digit (~13%, n = 2) or 1-digit hand (~6%, n = 1) formed on the lower arm (Figure 3; Table 2).

#### 3.2.2. Posterior-Posterior (Pos-Pos)

The skin on the anterior half of the upper arm was replaced with skin obtained from the posterior half of the contralateral upper arm, while keeping the proximodistal and dorsoventral axes of the skin unchanged (Figure 4). The operated limbs regenerated normally (~81%, 13/16) except for those cases (~19%, 3/16) in which a hand with supernumerary digits appeared (Figure 4; Table 2). In one case (~6%, n = 1), a hand appeared to have lost its dorsoventral pattern and the excessive digits were irregularly formed on the hand whose palm was missing. In the other cases (~13%, n = 2), a butterfly-like hand was formed in which two pairs of symmetrical hands and lower arms appeared to fuse horizontally along the midline. We never found any extra forearm formation from the limb.

#### 3.2.3. Ventral-Ventral (Ven-Ven)

The skin on the dorsal half of the upper arm was replaced with skin obtained from the ventral half of the contralateral upper arm while keeping the proximodistal and anteroposterior axes of the skin unchanged (Figure 5). All of the operated limbs regenerated normally (100%, n = 31) (Figure 5; Table 2).

#### 3.2.4. Dorsal-Dorsal (Dor-Dor)

The skin on the ventral half of the upper arm was replaced with skin obtained from the dorsal half of the contralateral upper arm while keeping the proximodistal and anteroposterior axes of the skin unchanged (Figure 6). In this operation, most of the operated limbs regenerated normally (~76%, 19/25) except for several cases (Figure 6; Table 2). In one case, as observed in the An-An pattern of the operation (Figure 3D), a lower arm with a 5-digit hand appeared (~4%, n = 1). In the other cases, like a butterfly-like hand observed in the Pos-Pos pattern of the operation (Figure 4G–I), a limb with a supernumerary digit hand appeared (~20%, n = 5). However, unlike the butterfly-like hand, these supernumerary digit hands looked as if two hands with the normal axial pattern partially overlapped and fused together. The skeletal stain revealed that additional digits appeared on the anterior side of the palm (Figure 6H).

#### 3.2.5. Flank/Tail Skin Graft

These findings suggest that the interaction of the adjacent skin is not essential for the regeneration of the patterned limbs. To confirm this conclusion, we further investigated the effect of skin from different parts of the body on limb regeneration (Figure 7). As anticipated, when the skin of half of the upper arm was replaced with skin of the flank of the same individual, all the operated limbs regenerated normally (An–Flank, Flank–Pos, Flank–Ven, Dor–Flank; n = 3 each; Table 2). When the skin of the ventral half of the upper arm was replaced with the skin of the tail of the same individual, the limbs regenerated normally in two of the three newts examined (Table 2); in the other one, a 5-digit hand appeared (Figure 7F,G; Table 2) as with An-An (Figure 3D) and Dor-Dor (Figure 6E). These results suggest that the axial pattern of the blastema is normally formed regardless of the pattern or type (origin) of the skin around the stump.

#### 3.2.6. Control

Next, we amputated the upper arm along the gap between the host skin and the distal edge of the half skin immediately after placing it on the graft site. As a result, in either the An-An, Pos-Pos, Ven-Ven, or Dor-Dor patterns, all the limbs regenerated normally (n = 2 each, Table 2). These results rule out the possibility that the one-month interval between skin grafting and amputation allowed the grafted skin to adjust its original geometrical identity to the graft site.

In the sham operation in which the half skin on either the anterior, posterior, ventral, or dorsal side of the upper arm was grafted back to the original position immediately after it had been peeled off, all the limbs regenerated normally (n = 3 each; Table 2). These results suggest, as in 180° skin rotation, that the surgery itself is not responsible for the morphological abnormalities of the regenerated limb in the half skin graft operation, but rather due to some errors during the operation of translocating the skin.

## 4. Discussion

In amphibians, the skin at the stump is thought to be important for the axial pattern formation in regenerating limbs [2,3,8,11,12,13,14,15]. However, as was shown in this study, the skin manipulation did not have a serious impact on the axial patterning in adult newt forelimb regeneration. This suggests that the mesenchymal cells of the blastema and the epidermis surrounding the blastema, both derived from the skin, do not work autonomously in the axial patterning of the blastema relying on the memory of their original geometric identity. On the other hand, the morphological abnormalities such as supernumerary digits did occur, although less frequently. Interestingly, these abnormalities were related to the skin grafting pattern, and some abnormalities were reproducible (except for a hand with an additional 5th digit next to the 4th digit; Table 2). We propose here a hypothesis that the tissue involved in the axial patterning of the blastema may be subcutaneous, although the substance is not yet clear. These tissues must be accompanied by the cut skin to the graft site and must simultaneously be the tissues that are removed from the subcutaneous region when the skin is cut. It is possible that these operational errors occurred at low frequencies. Our hypothesis is as follows (Figure 8; also see Appendix A).

The tissues involved in the patterning of the regenerating forelimb are predicted to be located in a narrow area under the skin on the anterior and posterior sides of the ventral region of the upper arm. In contrast to humans, the forelimbs of the adult newt do not have typical subcutaneous tissue containing a layer of fat. Instead, the skin of adult newt is in loose contact with a thin connective tissue layer that surrounds the muscles, in which nerves and capillaries for blood and lymph fluid are embedded. In this study, though we carefully separated the skin from the connective tissue layer on the muscle, some of the connective tissue layer may have remained in the skin at a low rate. The Ventral Posterior Area (VPA) may be the origin of the mesenchymal cells of the blastema that are involved in the posteriorization of the blastema. These mesenchymal cells may constitute a region corresponding to the Zone of Polarizing Activity (ZPA) of the limb bud of the tetrapod embryo [19,20,21,22]. Mesenchymal cells in the ZPA are known to express Shh, a signaling molecule that governs posteriorization of the limb bud [19,20,21,22]. In fact, in adult newt blastema, the Shh-expressing region is localized posteriorly, as is the ZPA in limb buds [20]. The butterfly-like hand (with symmetrical digits) observed in the Pos-Pos pattern of the half skin graft operation (Figure 4G–I) corresponds to the result of implanting another ZPA on the anterior side of the limb bud [19,20]. Therefore, it is plausible that in the case where the butterfly-like hand appeared, the VPA that happened to accompany the posterior half of the skin during the skin removal process was grafted to the anterior side, resulting in a limb with dual VPAs (Appendix A).

In the blastema, the mesenchymal cells originating from the Ventral Anterior Area (VAA) may be involved not only in the anteriorization of the blastema but also in the ventralization of the blastema in cooperation with the mesenchymal cells originating from the VPA. For example, a phenotype observed in 180° skin rotation in which the regenerated limb was rotated 90° forward around the proximodistal axis and the digit alignment of the hand was reversed (Figure 2A–H) can be explained by 180° rotation of the VAA which happens to accompany the operation (Appendix A).

Other morphological abnormalities can also be explained by VPA and VAA, which can be independently defective or damaged during the grafting process, or carried with the grafted skin (for details, see Appendix A). However, the supernumerary digits observed in the Dor-Dor pattern of the half skin graft operation is thought to be due to a separation and displacement of the grafted skin (most likely along the anterior–posterior axis) during the amputation or in the period between the amputation and the blastema formation (Appendix A). The ventral side of the stump in the forelimb is more susceptible to physical damage because it has greater contact with the ground. In fact, in the Dor-Dor pattern of the half skin graft operation, the grafted region on the stump was sometimes wounded during rearing, damaging the blastema and almost stopping the regeneration (5 (17%) of 30 newts; these cases were not added to the results). The possibility that the supernumerary digits arose as a result of an uneven wound epithelium formed between the grafted skin and the host skin would be ruled out because the limb regenerated even when the flank or tail skin was grafted.

In limb bud formation in tetrapods, the epidermis covering the limb buds is required for the patterning along the proximodistal and dorsoventral axes, and the mesenchymal cells comprising the ZPA are required for the patterning along the anteroposterior axis [21,22]. Based on the results of this study, we predicted a region (VPA) in the normal limb as the origin of the cell cluster corresponding to the ZPA in the blastema. On the other hand, if, by analogy with the limb bud development, the epidermis of the blastema is also involved in the patterning of the proximodistal and dorsoventral axes, then the results of the present study predict that the epidermis specializes in the blastema formation independently of the geometrical identity of the skin from which the epidermis originates. In fact, it has been suggested that the specialization of the epidermis at the distal part of the blastemal, by the interaction with the regenerating nerve, is important for the pattern formation along the proximodistal axis [6,10]. Perhaps, for the dorsoventral axis as well, the epidermis surrounding the ventral side of the blastema may interact with the mesenchymal cells in the blastema, which are derived from the ventral subcutaneous area (VPA–VAA) assumed in this study.

However, we do not rule out the possibility that the skin adjusted its geometrical identity to the graft site as soon as it was placed at the graft site, so that the effects of skin manipulation were minimal. If this is the case, it is possible that the mesenchymal cells of the blastema and the epidermis surrounding the blastema, both of which originate from the skin of the stump, could work on the basis of their overwritten/replaced memory of the geometrical identity. In this case, however, the failure of the skin to adapt to the graft site over the one-month period must be considered as a cause of the morphological abnormalities associated with the skin manipulation, in addition to the loss/damage and contamination of the subcutaneous tissue at the graft site as discussed above. Importantly, this hypothesis also assumes that the interaction with the subcutaneous tissue is necessary for the skin to acquire or replace its geometrical identity.

In this study, we further attempted to determine which of the mesenchymal cells of the blastema that contributed to the patterning of the blastema originated from the skin or subcutaneous tissue by using a skinless limb model and a subcutaneous tissueless limb model (see Appendix A). Surprisingly, even under such severe situations in both models, the normal pattern of the limbs regenerated (n = 3 each). However, it was difficult to rule out the possibility that the dermal mesenchymal cells participated, although the tissues under the skin were obviously essential. Additionally, we conducted experiments to trace the dermal cells after amputation by grafting the skin stained with a tracer dye or the skin obtained from reporter-expressing transgenic newts. However, in the former experiment, the staining conditions still need to be optimized so that the cells remain fluorescent at least until the blastema stage (~45 days). In the latter experiment, the skin allograft between individuals (e.g., between wild type and transgenic individuals) at the adult stage was difficult because of immunological rejection [18]. Alternatively, we explored the responsible cells in subcutaneous area by tissue grafting. We separated tissues including nerves and capillaries from the area corresponding to the VPA and grafted them under the skin of the ventral anterior part of the contralateral upper arm. In the forelimbs of adult newts, a thick nerve (ulnar nerve) and blood vessels run along the VPA. Unexpectedly, however, no regeneration of the butterfly-like hand was observed (n = 10). One possibility is that the responsible cells are concentrated in a specific location or sparsely distributed in the connective tissue layer. Therefore, further investigation is needed.

In conclusion, our results suggest that the tissue under the skin, rather than the skin itself, in the intact limb is of primary importance in ensuring the normal axial pattern formation in adult newt limb regeneration. We propose a hypothesis that the important tissues may be located in VAA and VPA. However, we note that this study only focused on regeneration from the upper arm of the forelimb, since it is predictable that the mechanism of pattern formation in regeneration would not be the same between the upper and lower arms, or between the forelimb and hind limb [2]. In a future study, it will be important to identify the cells in the VAA and VPA, as well as to characterize them by gene expression profiling, so that we can explore the corresponding cells in humans. For this, we will need to develop new strategies that can overcome the problems that were encountered in this study. Tracking Shh cells by transgenesis is likely among the most promising of technologies.

## Figures and Tables

**Figure 1 biomedicines-09-01426-f001:**
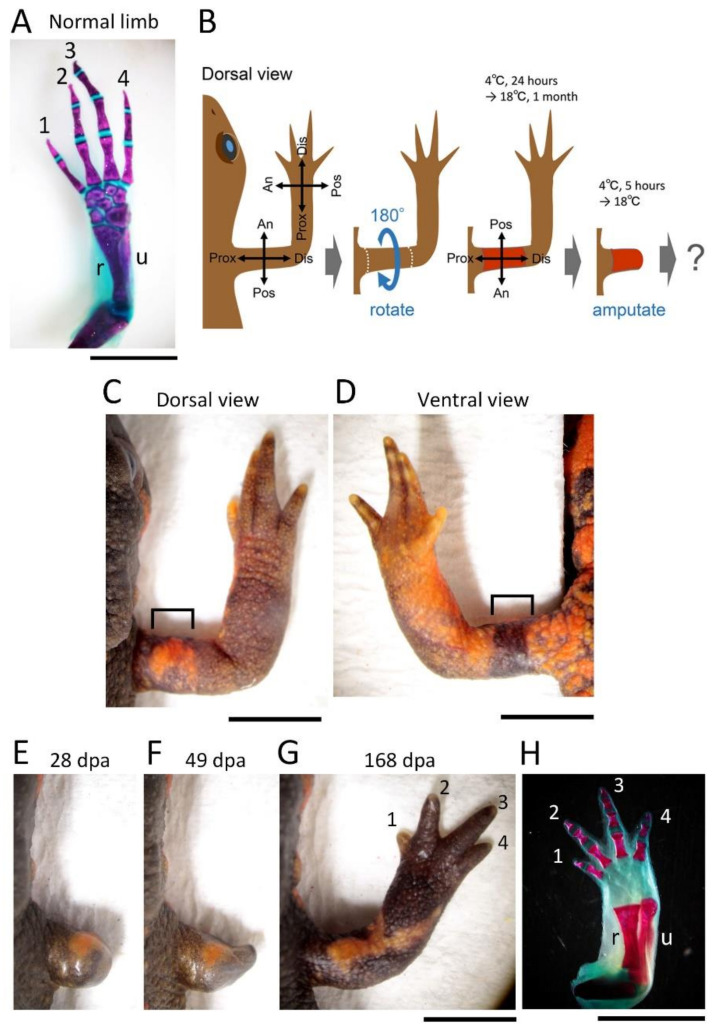
Normal regeneration of the limb with 180° rotated skin. (**A**) The skeleton of a normal forelimb. (**B**) Schematic showing the surgical process. (**C**–**H**) Representative showing normal regeneration of the operated limb (n = 17). (**C**,**D**) Dorsal and ventral views of the operated limb before amputation. Bracket: rotated skin. (**H**) The skeleton of the regenerated limb shown in (**G**). The skeleton in (**A**,**H**) as shown by Alizarin red (hard bone)-Alcian blue (cartilage) staining. Note that the cartilage of the regenerating limb is not always stained blue as shown in (**H**) (almost transparent). The number near the digit indicates digit number. dpa: day post amputation; u: ulna; r: radius. Scale bars: 5 mm.

**Figure 2 biomedicines-09-01426-f002:**
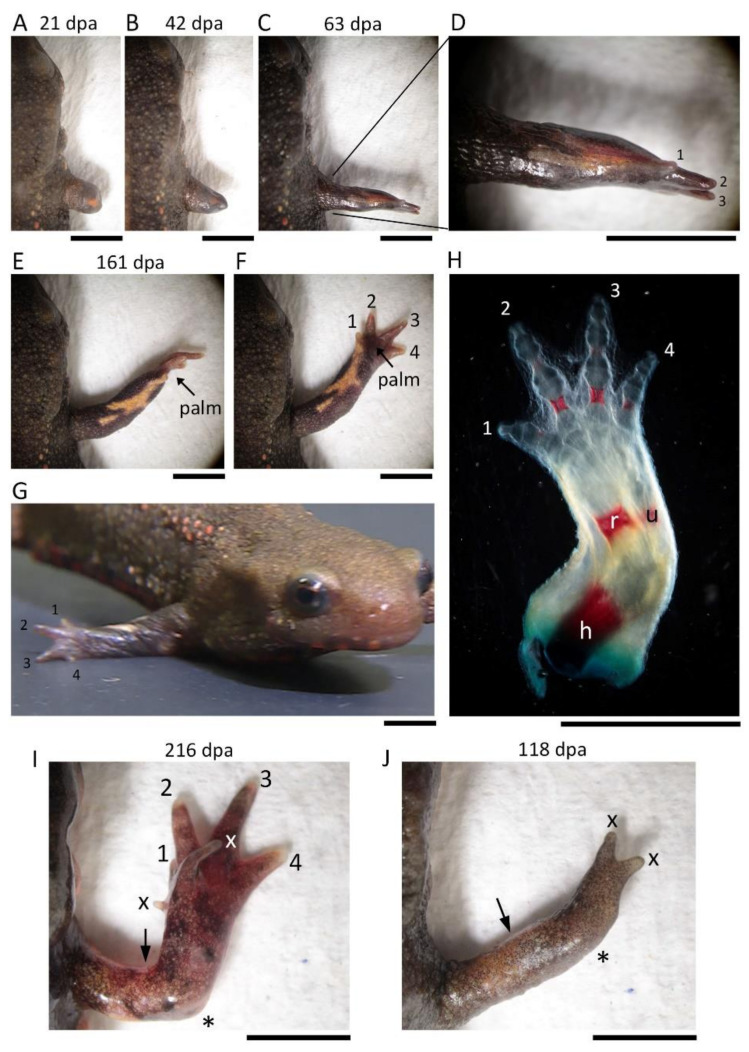
Abnormally regenerated limbs in 180° skin rotation. (**A**–**H**) Representative of a limb rotated 90° forward with a hand in which the order of the digits is reversed (n = 3). (**E**–**G**) Images of the same animal taken from different angles. (**H**) Skeleton of the regenerated limb shown in (**E**). Hard bone was stained red. (**I**) A limb with a hand with extra digits on the front side of the back of the hand (n = 1). (**J**) A limb with two digits at the top (n = 1). The number near the digit indicates digit number. X indicates the digit whose number could not be determined. The asterisk indicates the elbow. The arrow indicates the amputation position. dpa: day post amputation; u: ulna; r: radius. h: humerus. Scale bars: 5 mm.

**Figure 3 biomedicines-09-01426-f003:**
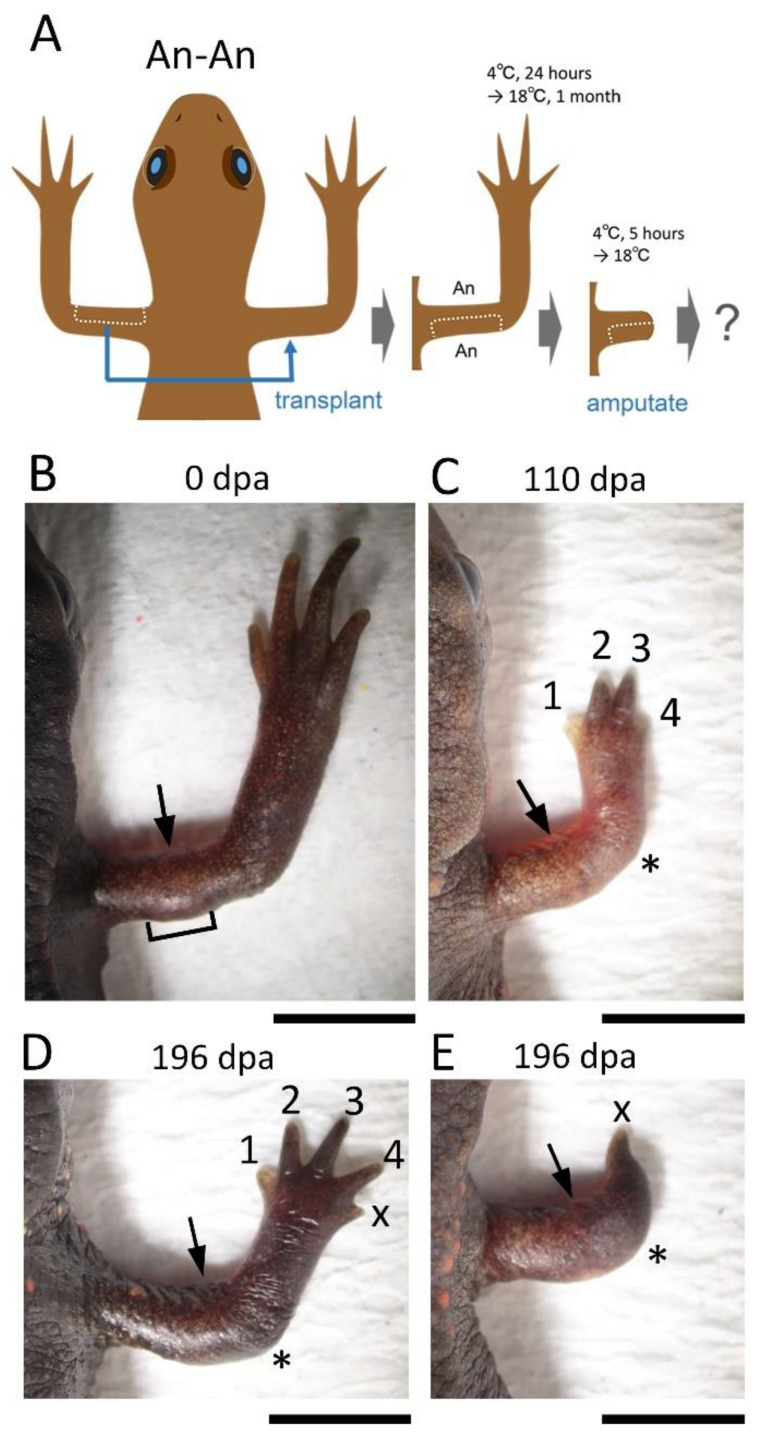
Effects of Anterior-Anterior (An-An) skin grafting on limb regeneration. (**A**) Schematic showing the surgical process. (**B**,**C**) Representative showing normal regeneration (n = 13). Bracket: grafted skin. (**D**,**E**) Rarely occurring abnormal limbs. (**D**) A limb with a 5-digit hand (n = 2). (**E**) A limb with a 1-digit hand at the tip (n = 1). The number near the digit indicates digit number. X indicates the digit whose number could not be determined. The asterisk indicates the elbow. The arrow indicates the amputation position. dpa: day post amputation. Scale bars: 5 mm.

**Figure 4 biomedicines-09-01426-f004:**
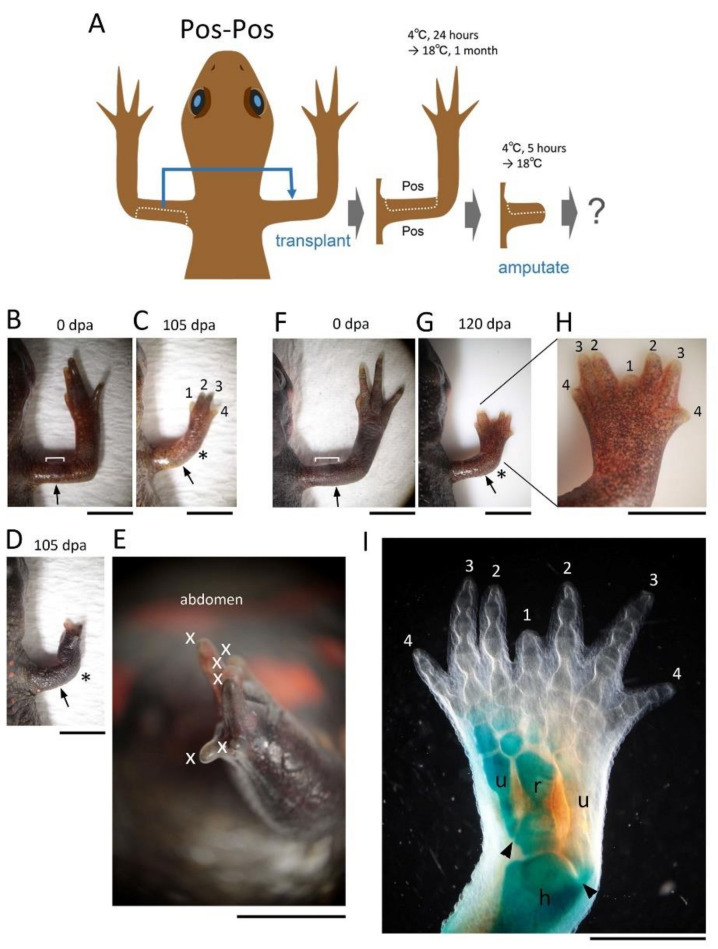
Effects of Posterior-Posterior (Pos-Pos) skin grafting on limb regeneration. (**A**) Schematic showing the surgical process. (**B**,**C**) Representative showing normal regeneration (n = 13). (**D**–**I**) Rarely occurring abnormal limbs. (**D**,**E**) A limb with excessive digits on a hand with impaired dorsoventral pattern (n = 1). (**E**) An enlarged image at a different angle of the limb shown in (**D**). The palm was not recognized. (**F**–**H**) Representative of a limb with a butterfly-like hand (n = 2). The bracket indicates the grafted skin. The number near the digit indicates digit number. X indicates the digit whose number could not be determined. The asterisk indicates the elbow. The arrow indicates the amputation position. (**I**) The skeleton of the limb shown in (**H**). Two pairs of symmetrical hands and lower arms appeared to fuse horizontally along the midline (along the radius and first digit). In this sample, the cartilage was partially stained blue. The arrowhead in (**H**) indicates a joint made at the proximal end of the ulna. The posterior ulna articulated to the humerus as normal but the anterior ulna ended with the radius, indicating that the anterior half of the regenerated limb was not completely mirrored. dpa: day post amputation; u: ulna; r: radius; h: humerus. Scale bars: 4 mm (**B**–**D**,**F**–**H**); 3 mm (**E**); 2 mm (**I**).

**Figure 5 biomedicines-09-01426-f005:**
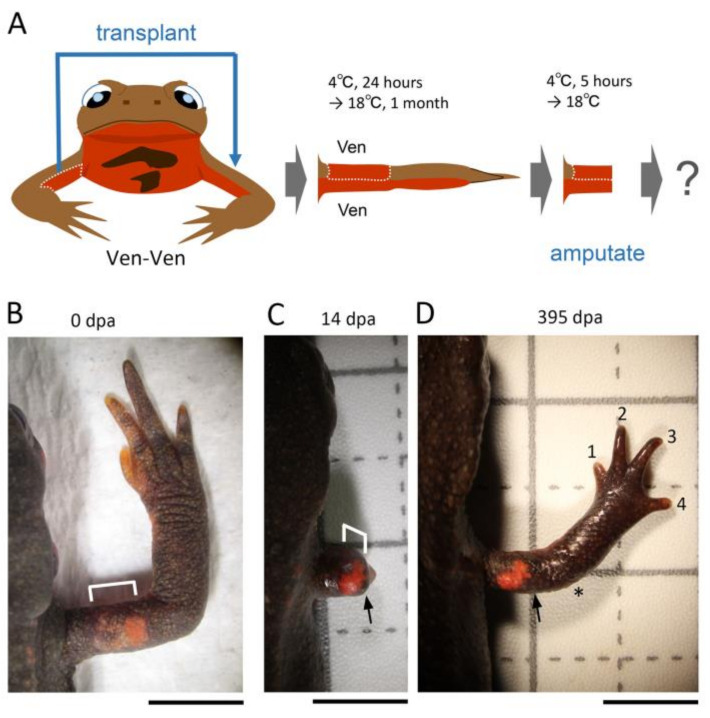
Effects of Ventral-Ventral (Ven-Ven) skin grafting on limb regeneration. (**A**) Schematic showing the surgical process. (**B**) A limb immediately before amputation. (**C**,**D**) Representative image showing normal regeneration (n = 31). The bracket indicates the grafted skin. The number near the digit indicates digit number. The asterisk indicates the elbow. The arrow indicates the amputation position. dpa: day post amputation. Scale bars: 5 mm.

**Figure 6 biomedicines-09-01426-f006:**
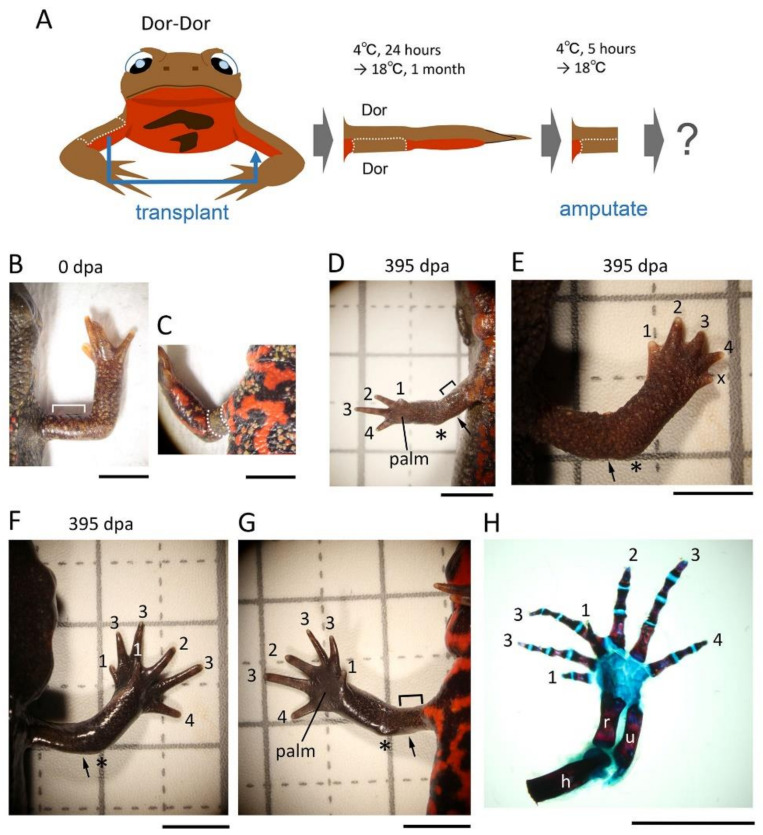
Effects of Dorsal-Dorsal (Dor-Dor) skin grafting on limb regeneration. (**A**) Schematic showing the surgical process. (**B**,**C**) A limb immediately before amputation. The image in (**C**) is a ventral view of the limb shown in **B**. Dotted lines indicate the border of the grafted skin. (**D**) Representative showing normal regeneration (ventral view; n = 19). (**E**) A limb with a 5-digit hand (dorsal view; n = 1). (**F**–**H**) Representative showing a limb with a supernumerary digit hand (n = 5). In this case, the two right hands seemed to partially overlap and fuse together. (**G**) is a ventral view of the limb shown in (**F**). (**H**) shows a dorsal view of the skeleton of the limb shown in (**F**). The number near the digit indicates digit number. X indicates the digit whose number could not be determined. The asterisk indicates the elbow. The arrow indicates the amputation position. dpa: day post amputation; u: ulna; r: radius; h: humerus. Scale bars: 5 mm.

**Figure 7 biomedicines-09-01426-f007:**
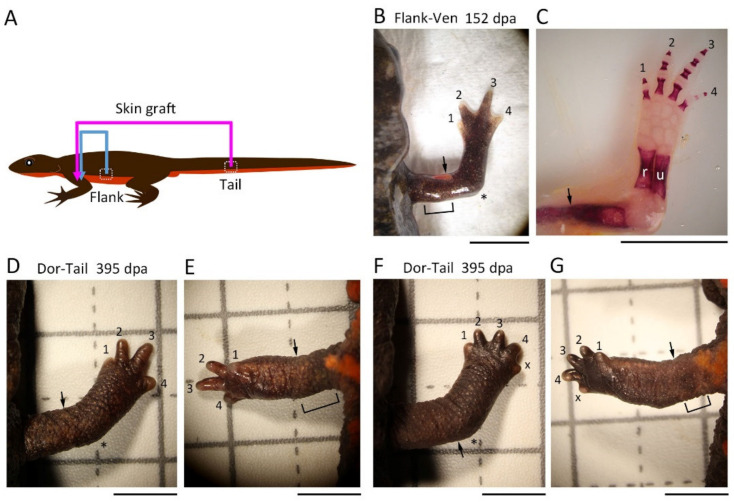
Effects of flank/tail skin grafting on limb regeneration. (**A**) Schematic showing the surgical process. (**B**) Representative showing normal regeneration in the Flank–Ven pattern of skin graft (n = 3). In this operation, a dorsal half skin was replaced with flank skin. (**C**) The skeleton of the limb shown in (**B**). (**D**) Representative showing normal regeneration in the Dor–Tail pattern of skin graft (dorsal view; n = 2). In this operation, a ventral half skin was replaced with tail skin. (**E**) Ventral view of the limb in (**D**). (**F**) A regenerating limb with a 5-digit hand in the Dor–Tail pattern of skin graft (dorsal view; n = 1). (**G**) Ventral view of the limb in (**F**). The number near the digit indicates digit number. X indicates the digit whose number could not be determined. The asterisk indicates the elbow. The arrow indicates the amputation position. dpa: day post amputation; u: ulna; r: radius; h: humerus. Scale bars: 5 mm (**B**,**D**–**G**); 1 cm (**C**).

**Figure 8 biomedicines-09-01426-f008:**
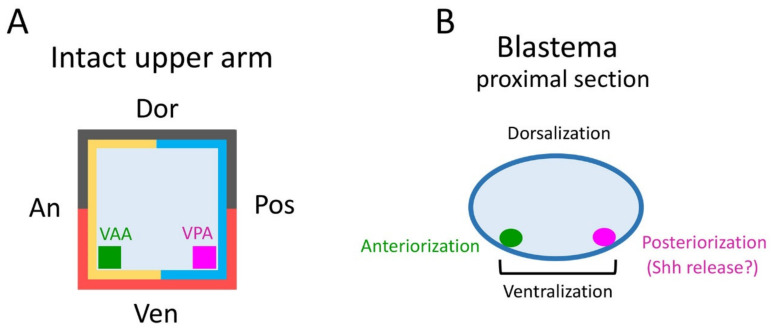
A hypothesis of subcutaneous area involved in the axial patterning of the blastema. (**A**) Schematic showing a cross section of an intact upper arm. Essential tissues that give rise to blastemal mesenchymal cells which work for axial patterning of the blastema may be localized under the ventral anterior and ventral posterior skin (green and magenta). VAA: Ventral Anterior Area. VPA: Ventral Posterior Area. (**B**) Schematic showing a cross section of a proximal part of the blastema. Green and magenta show blastemal cells originating from VAA and VPA, respectively. Magenta cells may play a role in posteriorization of the blastema by forming posterior digits. It may correspond to the Zone of Polarizing Activity (ZPA) in the limb bud, which secretes Shh. Green cells may work, together with magenta cells, to ventralize as well as anteriorize the blastema.

**Table 1 biomedicines-09-01426-t001:** Effects of 180° skin rotation on the axial pattern of the regenerating limb.

Skin Manipulation (Total Number)	Normal	Abnormal		
		90° Rotation with Digits in Reverse Order	Additional Digits on the Anterior Side of the Back of the Hand	2 Digits
180° rotation (n = 22)	17	3	1	1
Sham surgery (n = 3)	3	0	0	0
Skin removal (n = 3)	3	0	0	0

**Table 2 biomedicines-09-01426-t002:** Effects of half skin graft operation on the axial pattern of the regenerating limb.

Half Skin Graft Operation (Total Number)	Normal	Abnormal				
		Supernumerary Digits				Deficient Digits
		Irregular	Symmetry	Additional Digits on the Anterior Side of the Palm	Additional 5th Digit Next to 4th Digit	1 Digit
Anterior-Anterior (n = 16)	13	0	0	0	2	1
Posterior –Posterior (n = 16)	13	1	2	0	0	0
Ventral-Ventral (n = 31)	31	0	0	0	0	0
Dorsal-Dorsal (n = 25)	19	0	0	5	1	0
Flank skin graft (n = 3 each)	3 each	0	0	0	0	0
Tail skin graft (Dorsal-Tail, n = 3)	2	0	0	0	1	0
Immediate amputation (n = 2 each)	2 each	0	0	0	0	0
Sham surgery (n = 3 each)	3 each	0	0	0	0	0

## Data Availability

All data used in this study are available from the corresponding author upon reasonable request.

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
