# Peer review of "Reviewing the Effects of Skin Manipulations on Adult Newt Limb Regeneration: Implications for the Subcutaneous Origin of Axial Pattern Formation"

_biomedicines, 2021, doi:10.3390/biomedicines9101426_

Round 1

Reviewer 1 Report

This is an interesting study examining the role of upper arm skin and blastema formation and pattern in adult newts. 

This study concludes that normal regeneration pattern forms regardless of limb skin origin. 

Previous studies in other salamanders suggest that the skin, dermal fibroblasts in particular, play a critical role in limb regenerate patterning. 

So, in this way, this study is in conflict with those. 

Questions I had reading this manuscript: 

Following grafting and immediate amputation, how do you know that the grafted skin was not shed or lost. 

Skin removal option, why were you surprised that the limb had to recover its epithelia before regeneration? The role of wound epithelia has been well demonstrated. 

How do you know that the 1-2 mm of skin remaining from your graft is still present during blastema formation and that during dedifferentiation and blastema formation the limb doesn't just resorb your grafted tissue back to the host level? The examples you share in your figures have some distinct skin in most cases but not all. Was the persistence of grafted skin evident in all your animals? Is this important to mention/consider. 

Labeling the skin is critical to demonstrating this. Grafting between transgenic siblings would theoretically resolve tissue rejection. 

The supernumerary stray digits could be the result of uneven wound epithelia formation between your grafts and host tissues. 

No mention that the "butterfly hand" actually has extra zeugopedial structures. This looks like the result you might have seen in an axolotl. 

I am left wondering if you would have had the same result with younger animals?

How did you control for grafting underlying tissues, no discussion of examining the graft for contaminating deeper tissue is mentioned. 

Did you consider purposely grafting deep tissue to examine if this is what resulted in your supernumerary zeugopod/digit cases? 

The formation of supers could also be the result of post surgery, post amputation trauma. You state that "(such cases were not added to the results)" but don't state how many times this happened. I'm having trouble visualizing your crumpled paper protocol and how long the animals were kept constrained in this. You also state that "we did not count the animals that obviously suffered damage on the grafted skin from ampuation or on the blastema during rearing". How often and how many cases did this represent? You mention that the animals were kept out of water to prevent infection. Is regeneration rate slowed in this case? 

The references are appropriate. 

The writing is clear and easy to follow.

The figures are well prepared and useful. 

The bone staining is great but the cartilage staining is not consistent. But, this is okay as you can "see" what you need to see. 

More reflection on how this differs with other previous studies in the conclusion would strengthen this paper and make it more useful going forward. Moreover, clearer plans for future studies would also enhance this paper. 

Newt skin is tough'/thick in comparison to axolotl larvae and adults, grafting tissues to newts is also tough. As a result, amputations immediately after grafting may be problematic - especially if close attention to graft retention is not mentioned. It might be worth clarifying this for your audience. 

Author Response

To Reviewer #1:

First of all, we would like to thank the reviewer for valuable suggestions and comments. Here we attempted to respond to the comments.

The reviewer is mainly concerned about how well the grafted skin was preserved. To clarify this point, we modified the contents of sections, 2.3. Skin manipulation and 2.4. Limb amputation, of Materials and Methods as follows:

Page 3. Line 44- Page 4. Line 2. The operated animals were placed in a Tupperware box (W: 14.1 cm D: 21.4 cm, H: 4 cm; one animal per box) containing crumpled pieces of half-dried paper towel (Elleair Prowipe, Soft High Towel, Unbleached, 4P) and allowed to recover at 4 °C for 24 h. During the incubation at 4 °C, the animals were under anesthesia and did not move, so the grafted skin adhered to the limb without shedding. Then, the animals were reared in the same box at 18 °C for one month during which the gap of skin closed and the grafted skin became firmly attached to the limb.

Page 4. Line 17-22. The grafted skin was easily distinguished from the host skin by its characteristic color pattern. We frequently observed the grafted skin under a dissecting microscope during the process of rearing animals after skin grafting or amputation. The grafted skin maintained its characteristics on the host limb for more than a year. However, if the grafted skin was detached or lost, the animal was excluded from the experiment (the case was less than 5% on average).

Page 4. Line 25-28. In the case of the immediate amputation option, the upper arm was amputated along the gap between the host skin and the distal margin of the half skin immediately after the half skin was placed on the limb (the half skin was not cut again to prevent its detachment or loss).

Q1. Following grafting and immediate amputation, how do you know that the grafted skin was not shed or lost.

A1. As we mentioned above, the grafted skin was easily distinguished from the host skin by its characteristic color pattern. We frequently observed the grafted skin under a dissecting microscope during the process of rearing animals after skin grafting or amputation.

Q2. Skin removal option, why were you surprised that the limb had to recover its epithelia before regeneration? The role of wound epithelia has been well demonstrated.

A1. Thank you for the suggestion. We mentioned the reason in Page 7, Line 3-6, as follows: Moreover, this shows that the position of the skin from which the wound epidermis originates does not affect the positional identity or axial patterning of the blastema, because the wound edge of the skin is located more proximal to the tip of the stump.

Q3. How do you know that the 1-2 mm of skin remaining from your graft is still present during blastema formation and that during dedifferentiation and blastema formation the limb doesn't just resorb your grafted tissue back to the host level? The examples you share in your figures have some distinct skin in most cases but not all. Was the persistence of grafted skin evident in all your animals? Is this important to mention/consider.

A3. We agree with the reviewer’s opinion.  As mentioned above, we explained the persistence of grafted skin in Materials and Methods as follows: The grafted skin maintained its characteristics on the host limb for more than a year.

Q4. Labeling the skin is critical to demonstrating this. Grafting between transgenic siblings would theoretically resolve tissue rejection.

A4. As we noted in Page 4, Line 14-16, skin allograft between adult individuals is impossible because the grafted skin dissolves and disappears, probably due to immunological rejection by the host [18].

We have already confirmed this phenomenon by skin cell tracking using transgenic newts (we are now studying the mechanism of immunological rejection). Therefore, in this study, we adopted skin autografting, in which we never observed such immunological rejection. As mentioned above, the grafted skin is easily distinguished from the host skin by its characteristic color pattern. We frequently observed the grafted skin under a dissecting microscope during the process of rearing animals after skin grafting or amputation. The grafted skin maintained its characteristics on the host limb for more than a year.

Q5. The supernumerary stray digits could be the result of uneven wound epithelia formation between your grafts and host tissues. 

A5. The reviewer concerns about the cause of supernumerary stray digits observed in the DD pattern of half skin graft operation. We mentioned our statement in Page 15, Line 13-16 as follows: The possibility that the supernumerary digits arose as a result of an uneven wound epithelium formed between the grafted skin and the host skin would be ruled out, because the limb regenerated even when the flank or tail skin was grafted.

Q6. No mention that the "butterfly hand" actually has extra zeugopedial structures. This looks like the result you might have seen in an axolotl. 

A6. As shown in Figure 4, unlike an accessory limb model in axolotl, the limb of Pos-Pos pattern never had extra zeugopedial structures, or extra forearms. We mentioned this result in Page 9, Line 18 as follows: We never found any extra forearm formation from the limb.

Q7. I am left wondering if you would have had the same result with younger animals?

A7. No, we have not done the same experiments with younger newts.

Q8. How did you control for grafting underlying tissues, no discussion of examining the graft for contaminating deeper tissue is mentioned. Did you consider purposely grafting deep tissue to examine if this is what resulted in your supernumerary zeugopod/digit cases?

A8. We mentioned the following sentence in Page 14, Line 20-24:  The skin of adult newt is in loose contact with a thin connective tissue layer that surrounds the muscles, in which nerves and capillaries for blood and lymph fluid are embedded. In this study, though we carefully separated the skin from the connective tissue layer on the muscle, some of the connective tissue layer may have remained in the skin at a low rate.
    In addition, we mentioned our observations and statement in discussion part (Page 16, Line 3-11) as follows:  Alternatively, we explored the responsible cells in subcutaneous area by tissue grafting: we separated tissues including nerves and capillaries from the area corresponding to the VPA and grafted them under the skin of the ventral anterior part of the contralateral upper arm. In the forelimbs of adult newts, a thick nerve (ulnar nerve) and blood vessels run along the VPA. Unexpectedly, however, no regeneration of the butterfly-like hand was observed (n=10). One possibility is that the responsible cells are concentrated in a specific location or sparsely distributed in the connective tissue layer. Further investigation is needed.

Q9. The formation of supers could also be the result of post surgery, post amputation trauma. You state that "(such cases were not added to the results)" but don't state how many times this happened. I'm having trouble visualizing your crumpled paper protocol and how long the animals were kept constrained in this. You also state that "we did not count the animals that obviously suffered damage on the grafted skin from ampuation or on the blastema during rearing". How often and how many cases did this represent? You mention that the animals were kept out of water to prevent infection. Is regeneration rate slowed in this case? 

A9. On average, less than 5% of the grafted skin detached or was lost (none of the 20 times). However, in the DD pattern of half skin grafting, the grafted skin was damaged relatively frequently (17%). We mentioned these information as follows:

Page 4, Line 20-22. However, if the grafted skin was detached or lost, the animal was excluded from the experiment (the case was less than 5% on average).

Page 4, Line 42-44. Note that in this study we did not count the animals that obviously suffered damage on the grafted skin from amputation or on the blastema during rearing (the case was less than 5% on average).

Page 15, Line 12-14. In fact, in the Dor-Dor pattern of half-skin graft operation, the grafted region on the stump was sometimes wounded during rearing, damaging the blastema and almost stopping regeneration (5 (17%) of 30 newts; these cases were not added to the results).

Regarding the rearing condition, the adult newts, C. pyrrhogaster, spend most of their time on land, like salamanders living on the forest floor, except during the breeding season. If you injure them, it is the best to use a moist paper towel as a moss and keep them on it. In the water, the survival rate of individuals decreases and regeneration becomes abnormal or stops. Our model is not a larva, but is fully adapted to the terrestrial environment.

Q10. More reflection on how this differs with other previous studies in the conclusion would strengthen this paper and make it more useful going forward. Moreover, clearer plans for future studies would also enhance this paper. 

A10. We agree with the reviewer’s opinion. However, it would be difficult to discuss about differences from previous studies because there is only limited information in adult newts. Besides, direct comparisons between adult newts and larval amphibians would also be difficult because as we mentioned in Introduction, the cellular participants and the mechanism of limb regeneration in larval amphibians would not all be the same as those in adult newts. Our focus is to understand strategies of limb regeneration employed by terrestrial tetrapods. As the reviewer suggested, we need more information about adult newt limb regeneration.

Currently, we are labelling Shh+ cells in blastema by transgenesis to track those cells during limb regeneration. In this experiment, we want to identify cell types which originate from Shh+ blastemal cells. If we amputate the regenerated limb again, we could find whether these cells give rise to Shh+ blastemal cells again. However, we would like to avoid making this project public in this paper, as it is being carried out by different authors and grants. Instead, we mentioned in Page 16, Line 21-21 just as follows:  Tracking Shh cells by transgenesis is probably one of the most promising technologies.

Q11. Newt skin is tough'/thick in comparison to axolotl larvae and adults, grafting tissues to newts is also tough. As a result, amputations immediately after grafting may be problematic - especially if close attention to graft retention is not mentioned. It might be worth clarifying this for your audience. 

A11. As mentioned above, we have clarified these points in the text.

Reviewer 2 Report

The manuscript is presenting various adult newt amputation models and skin manipulations to study the anterior limb regeneration process, with the aim of gaining an insight in a future human therapy. The work methods are sound, the data and methods are described very clearly, the figures and their captions are detailed and precise. Overall, the work was very interesting to read, considering the various hypothesis proposed by the authors.  

How are the location within the newts' body from which the skin pieces were taken for the graft decided? Are the skin operation reported in Table 2 the ones normally followed for the adult newts model?

Some infos on the different skin tissues' features, for example in terms of cell population, layered architecture, and mechanical properties, might be interesting for the reader. This would also clarify why so many combination of half-skin grafts have been considered in the study.

The skeleton staining is very interesting. A part from the bone fusion, is there any other conclusion that can be drawn from this characterization? (e.g. length of the regenerated bones, tissue density, etc...)

Regarding the digit numbers in supernumerary situation, how are they designated? For example, in figure 6H, three digits are labelled with the number 3 and two digits are labelled with the number 1. How this can be assigned in such an univocal way?

In the discussion the author are mentioning the Shh-expressing region. Would it be possible to assess the presence of Shh via immunofluorescence staining or it is rather complicated to perform (as commented in page 16 lines 39-42? Would this be an additional piece of info to support your hypothesis, to be acquired after animal sacrifice and tissue harvesting?

Very minor typos noticed are the following:

  • page 4 line 43: "alcian blue" is repeated twice
  • page 5 line 3: extra space to remove
  • page 5 line 17: correct the title of the paragraph
  • page 6 line 16: the sentence is not very clear. What do you mean with the expression "the skin was grafted back to the original position immediately after it peeled off"? In this case the skin was not used for the graft?
  • Figure 4D: correct "dop"
  • Figure S6: G is used twice to label different images

Author Response

To Reviewer #2:

First of all, we would like to thank the reviewer for valuable suggestions and comments. Here we attempted to respond to the comments.

Q1. How are the location within the newts' body from which the skin pieces were taken for the graft decided? Are the skin operation reported in Table 2 the ones normally followed for the adult newts model?

A1. As explained in the Introduction, these methods of skin grafting have been applied to larval or paedomorphic amphibians such as the axolotl. However, as for adult newts, there are only limited information. Therefore, following the previous studies in other larval amphibians, we developed the procedures for the adult newt C. pyrrhogaster by ourselves.

Q2. Some infos on the different skin tissues' features, for example in terms of cell population, layered architecture, and mechanical properties, might be interesting for the reader. This would also clarify why so many combination of half-skin grafts have been considered in the study.

A2. Our purpose in this study is to know whether the geometrical identity of the skin from which the wound epidermis originates influences the positional identity or axial patterning of the blastema. That is why we tested various combinations of half-skin grafts. However, unexpectedly, the skin did not influence axial patterning of the blastema. Therefore, we did not mention detailed information on the structure of the skin. In fact, we are currently preparing a paper that fully describes the skin regeneration process and skin structure of this newt. However, due to different authors and grants, we would like to avoid publishing this content in this paper.

Q3. The skeleton staining is very interesting. A part from the bone fusion, is there any other conclusion that can be drawn from this characterization? (e.g. length of the regenerated bones, tissue density, etc...)

A3. The differences in the length of bone and staining intensity between samples may be due to the difference in the degree of tissue maturation. Uneven staining may also be included. In this study, we mainly focused on the anteropoterior and dorsoventral patterns of regenerated limbs. Therefore, we reared animals until the morphological proportions of the limbs became stable. In fact, after the bone staining of limb samples, we found a varied degree of tissue maturation (or staining intensity). However, this did not affect the results. Therefore, in this study, regardless of the staining intensity, we characterized regenerated limbs by counting the number of cartilages/bones and joints as well as considering their relative length and positions in comparison with intact limbs.

Q4. Regarding the digit numbers in supernumerary situation, how are they designated? For example, in figure 6H, three digits are labelled with the number 3 and two digits are labelled with the number 1. How this can be assigned in such an univocal way?

A4. We identified digits by counting the number of cartilages/bones or joints (digit 1, 2; digit 2, 3; digit 3, 4; digit 4; 3). In case of the digit 2 and 3 both of which have 3 cartilages/bones or joints, we also considered the relative length of digits and their position. We mentioned these explanations in Materials and Methods part (Page 5, Line 17-19) as follows: The digits were identified by counting the number of cartilages/bones or joints (digit 1, 2; digit 2, 3; digit 3, 4; digit 4, 3). For the second and third digits, which have three cartilages /bones or joints, their relative length and position were also taken into account.

Q5. In the discussion the author are mentioning the Shh-expressing region. Would it be possible to assess the presence of Shh via immunofluorescence staining or it is rather complicated to perform (as commented in page 16 lines 39-42? Would this be an additional piece of info to support your hypothesis, to be acquired after animal sacrifice and tissue harvesting?

A5. As the reviewer commented, we have already created antibody against the newt Shh by ourselves (Shh antibody for the newt is not commercially available). However, Shh+ cells have not been detected in subcutaneous area. In parallel, we are labelling Shh+ cells in blastema by transgenesis to track those cells during limb regeneration. In this experiment, we want to identify cell types which originate from Shh+ blastemal cells. If we amputate the regenerated limb again, we could find whether these cells give rise to Shh+ blastemal cells again.

However, we would like to avoid making this project public in this paper, as it is being carried out by different authors and grants. Instead, we mentioned in Page 16, Line 21-21 just as follows:  Tracking Shh cells by transgenesis is probably one of the most promising technologies.

Q6. page 4 line 43: "alcian blue" is repeated twice

Q7. page 5 line 3: extra space to remove

Q8. page 5 line 17: correct the title of the paragraph

Q9. page 6 line 16: the sentence is not very clear. What do you mean with the expression "the skin was grafted back to the original position immediately after it peeled off"? In this case the skin was not used for the graft?

Q10. Figure 4D: correct "dop"

Q11. Figure S6: G is used twice to label different images

We would like to thank the reviewer for their careful corrections. We have corrected all these errors. As to Q9, the sentence has been changed to ‘the excised skin was grafted back to its original position immediately after the excision’. In this case, the skin was used.